# A framework for the risk prediction of avian influenza occurrence: An Indonesian case study

**Samira Yousefinaghani[1], Rozita Dara[1]***, Zvonimir Poljak[2], Fei Song[1], Shayan Sharif[3]**

**1** School of Computer Science, University of Guelph, Guelph, Ontario, Canada, **2** Department of Population Medicine, Ontario Veterinary College, University of Guelph, Guelph, Ontario, Canada, **3** Department of Pathobiology, University of Guelph, Guelph, Ontario, Canada

* drozita@uoguelph.ca

## Abstract

Avian influenza viruses can cause economically devastating diseases in poultry and have the potential for zoonotic transmission. To mitigate the consequences of avian influenza, disease prediction systems have become increasingly important. In this study, we have proposed a framework for the prediction of the occurrence and spread of avian influenza events in a geographical area. The application of the proposed framework was examined in an Indonesian case study. An extensive list of historical data sources containing disease predictors and target variables was used to build spatiotemporal and transactional datasets. To combine disparate sources, data rows were scaled to a temporal scale of 1-week and a spatial scale of 1-degree × 1-degree cells. Given the constructed datasets, underlying patterns in the form of rules explaining the risk of occurrence and spread of avian influenza were discovered. The created rules were combined and ordered based on their importance and then stored in a knowledge base. The results suggested that the proposed framework could act as a tool to gain a broad understanding of the drivers of avian influenza epidemics and may facilitate the prediction of future disease events.

## Introduction

Avian Influenza (AI) disease is caused by influenza type A viruses, which can infect domestic poultry, wild birds and mammalian species, including humans. Despite researchers' efforts to eradicate and control this disease, it has continuously caused significant losses to poultry and has threatened human lives. To mitigate the impact of AI outbreaks, it is necessary to understand the extent to which different risk factors and their interactions contribute to the introduction and spread of outbreaks. To date, an extensive array of studies have reported on spatiotemporal surveillance and control of AI using approaches, including logistic regression, boosted regression tree, cluster analysis and maximum entropy in different geographical scales. Studies have mapped the distribution of risk [1, 2] and identified important risk factors for disease occurrence [3, 4]. Some studies have performed spatiotemporal surveillance on a country scale such as those in Bangladesh [5], China [6], Indonesia [7], India [8], Thailand [1]

**Funding:** This work was funded by Egg Farmers of Canada, Chicken Farmers of Saskatchewan, and the Canadian Poultry Research Council. This research is supported in part by the University of Guelph's Food from Thought initiative, thanks to funding from the Canada First Research Excellence Fund.

**Competing interests:** The authors have declared that no competing interests exist.

and Vietnam [3] while others have focused on regional [9] or global [10] scales. Such disease risk-profiling approaches could assist in understanding the predictors of disease occurrence and preparing for future events.

Environmental conditions [11, 12], waterfowl [10, 13, 14], poultry farming and trading activities [15, 16], agricultural activities [17] and land cover [10, 18] are identified as major factors of introduction and dispersion of AI occurrence.

The impact of environmental factors and climate change on the spread and geographical distribution of AI outbreaks is documented in the literature [9, 19, 20]. For example, annual precipitation is introduced as an important predictive variable for the risk of highly pathogenic avian influenza (HPAI) in China [19]. Also, in the Middle East, the precipitation in the warmest quarter of a year is positively connected with HPAI H5N1 outbreaks [9]. In contrast, in Europe [20] and Bangladesh [4, 5], precipitation is negatively associated with H5N1 outbreaks in wild birds and poultry. Another important factor is the temperature that is positively associated with H5N1 outbreaks in wild birds in Europe [20] while an opposite pattern is found for poultry in Bangladesh [4, 5].

Moreover, the role of waterfowl density in the distribution of AI outbreaks is highlighted in a number of studies. In Asia [2, 21], domestic waterfowl density appears to be an important risk factor for H5N1 occurrence in poultry. Moreover, a positive association between duck density and H5N1 occurrence is found in Vietnam, Thailand [1, 22], India [8] and global scale [15].

Similarly, poultry density is considered as one of the factors associated with AI outbreaks. A strong association between HPAI outbreaks and densities of chickens was found in California [16], the Middle East [9] and globally [15]. Also, poultry market density in China is considered an important predictor of the risk of AI H7N9. In another study, Henning *et al.* [3] found that medium poultry density is associated with the risk of H5N1 outbreaks in Vietnam. Contrary to aforementioned studies, Yupiana *et al.* [7] found a negative association between H5N1 outbreaks and poultry density.

The spatial distribution of H5N1 outbreaks and its transmission to various regions have been associated with wild bird flyways [10, 23]. Moreover, the introduction of H5N1 to poultry in Europe, Asia and Africa may be partly through wild bird migration [24]. However, in a number of studies [25–27], a limited or a negative association has been found between migratory waterfowl sites and outbreaks of H5N1.

Despite the efforts that have been made to determine the essential predictor variables and suitable areas of AI presence, there are still some research gaps that need to be filled.

## Motivation

The present paper is aimed to obtain insights on the prediction of AI events using an extensive spatiotemporal dataset. For this, we identified gaps in the existing research concerning applied predictor variables and methodologies.

Despite the highlighted importance of risk predictor variables mentioned earlier, the precision and completeness of explanatory data have received limited attention in the literature. For example, a global climate system [28] has been frequently used in AI surveillance studies [9, 11, 29, 30]. The WorldClim website provides monthly average amount of climatic variables for various spatial resolutions. Clearly, the system only provides historical information and the monthly average of climatic variables is a low temporal resolution. Moreover, the geographic distribution of wild migratory birds has not been included in some existing work [15, 21, 31].

The existing approaches for determining how AI events occur in a region usually rely on regression [2, 6, 8, 19, 32] or boosted regression tree models [8, 15, 17, 18, 25]. The existing

work has aimed to find the most important predictors of disease and then profile the risk of outbreaks. In the aforementioned studies, the average impact of individual risk factors on the output is assessed. However, the identification of subgroups with different risk profiles is overlooked. This can consequently, ignore important information and produce biased results [17, 33]. Analysis relying on only one or a few numbers of risk factors could ignore important information and produce biased results.

The application of rule-based prediction models has been limited to a few studies for Dengue [12], Depression [34] and Diabetes [35, 36]. We are aware of only one study which used rule-based models aimed at analyzing AI outbreaks [37]. Xu *et al.* [37] constructed a data cube model with OLAP (Online Analytical Processing) actions. Then, geographical and temporal insights into disease spread with various abstraction levels were extracted. Moreover, sequential pattern mining and association rule mining were applied to provide understandings of potential serial spread routes and linkage between outbreak sites [37]. The researchers in this study Xu *et al.* [37] used the disease occurrence data regardless of the importance of explanatory variables.

The present study exploits historical data sources to extract predictive patterns of AI. The approach used here is complementary to prior research with three main contributions: (1) It uses several data sources with high temporal and spatial resolutions. (2) It employs rule-discovery models rather than focusing on predictive regression models commonly used in the relevant existing studies. Rule-discovery models generate patterns that can link the risk of disease presence with subsets of risk factors. (3) It contributes to automation and transparency of predictions. Although transparency of predictions is essential in application areas of epidemiology [38], the transparency of surveillance systems and their outcomes has received less attention in the literature. Since the proposed framework used a collection of rules as a high-level description of data, the explanation capability of predictions can be enhanced. This means public health officials can find the reasons behind the predictions made by the system.

The proposed framework was implemented and tested for an Indonesian case study. Indonesia was selected as a case study as this country has had a high number of reporoted AI outbreaks over the years and, importantly, it provides accessible explanatory data sources. This framework can form a basic model for risk prediction of AI events.

## Methodology

The main goal here is to extract prediction patterns of AI occurrence from a set of disparate data sources. We designed, implemented and tested a framework with four main parts including data collection, data aggregation and pre-processing, data analysis, and prediction. An overview of the main framework is presented in Fig 1.

In the first step (Data Collection), independent variables were identified and their respective data sources were collected. These variables were identified by the help of subject matter experts and from relevant literature. In the next step (Data Aggregation and Pre-processing), a relational database containing time and geographic information along with several covariates and outcome variables was designed. In the third step (Data Analysis), we applied rule discovery algorithms to the labelled dataset (training dataset) and extracted hidden patterns. These patterns indicated which combination of risk factors had led to high or low risk of disease occurrence and what linkage between event sites had been observed. Moreover, an experiment was conducted to evaluate the performance of predictions. In the last step (Prediction), end-users can communicate with the system through a user interface. Here, the user interface can include mapping and monitoring of the risk, given a current spatiotemporal dataset.

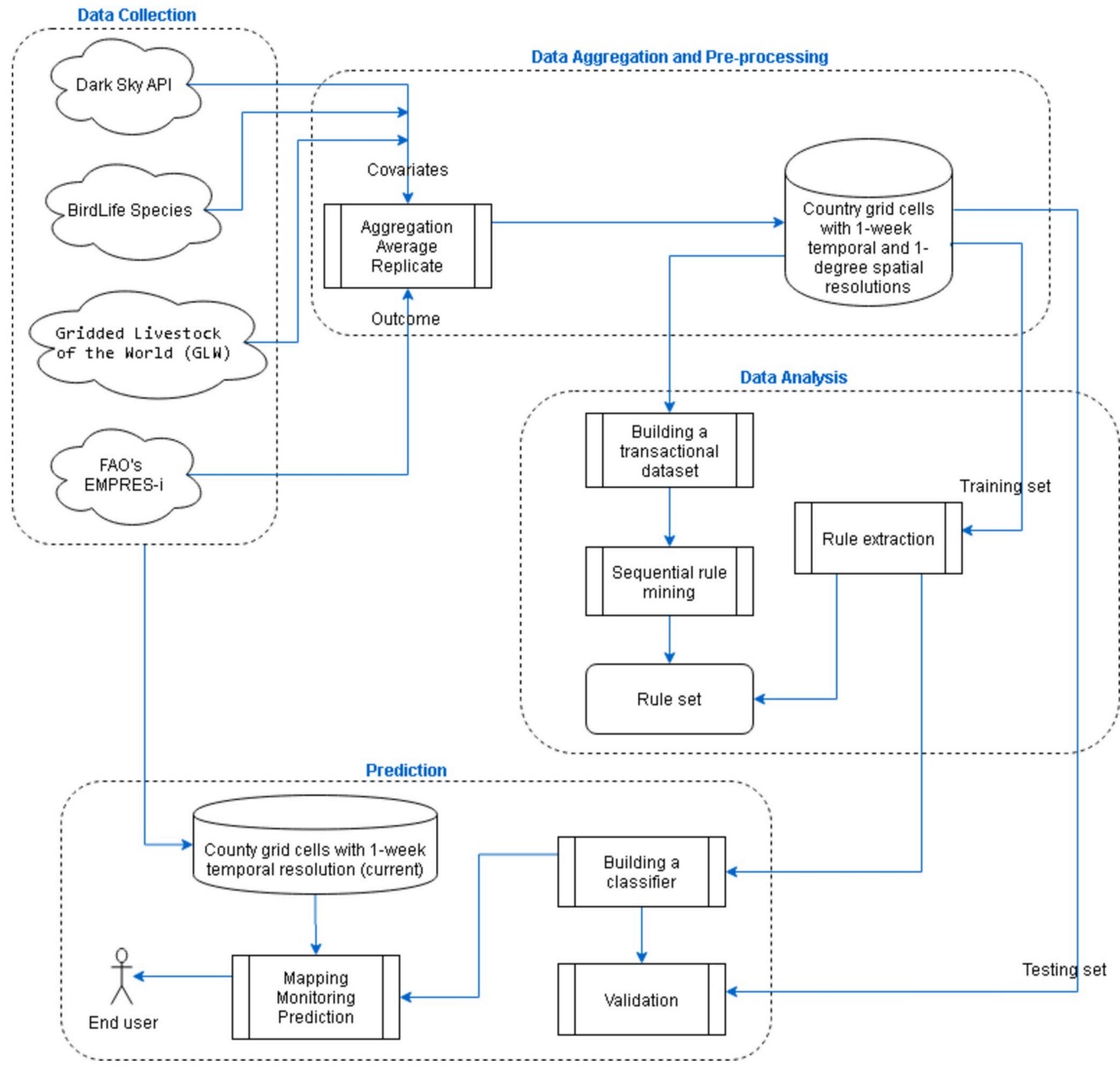

**Fig 1. Overall framework.**

## Data collection

A number of data sources have been downloaded and stored in their respective tables in a relational database that is visualized in Fig 2. Table 1 presents a summary of data sources used in the database construction. These data sources included climatic variables, geographical distribution of migratory bird species, distribution of poultry and AI historical records. More detail information on data tables is given in S1–S3 Tables in S1 File.

The risk factors obtained from these sources have been shown in the previous studies to correlate with AI outbreaks [12, 15, 16]. A list of risk factors used in the study along with their

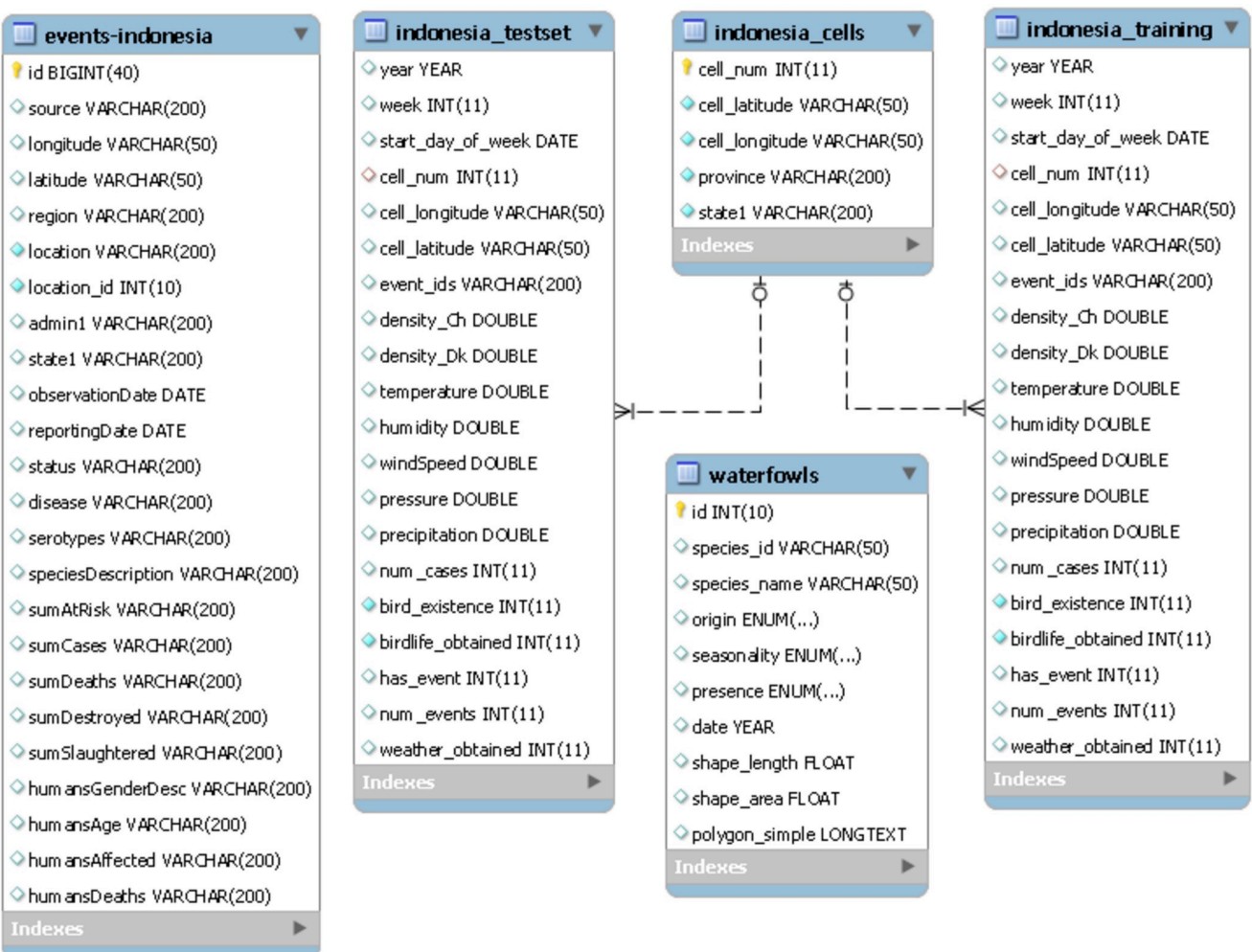

**Fig 2. Database schematic.**

**Table 1. Sources of data.**

| Data Source | Description |
|---|---|
| Dark Sky API | The API offers several climatic variables including temperature, humidity and wind speed. We automatically collected the variables that have been frequently used as risk factors of AI. The 'Time Machine Requests' API offered by Dark Sky [39] was used to retrieve weather information given latitude, longitude and time parameters. |
| BirdLife-species | The data provides geographic extents of species distribution ranges and is available in the Environmental Systems Research Institute (ESRI) Geodatabase formats [40]. |
| Gridded Livestock of the World (GLW3) | Food and Agriculture Organization (FAO) has developed the GLW3, in which the global distribution of chickens and ducks in 2010 is expressed by the total number of birds per pixel (5 minutes of arc) [41]. |
| EMPRES-i | FAO's Emergency Prevention System (EMPRES) offers a web-based application in order to facilitate the organization and access to disease data in various geographical scales which supports veterinary services [42]. |

**Table 2. Attributes.**

| Attribute | Type | Unit | Resolution |
|---|---|---|---|
| temperature | numerical | Fahrenheit | point |
| precipitation | numerical | millimetre | point |
| relative humidity | numerical | between 0 and 1 | point |
| wind speed | numerical | miles per hour | point |
| pressure | numerical | sea-level air pressure in millibars | point |
| chicken density | numerical | density | 5-minute arc |
| duck density | numerical | density | 5-minute arc |
| waterfowl | numerical | Boolean | point |

type, unit and resolution is shown in Table 2. These attributes are also visualized in 'indonesia_training' and 'indonesia_testset' data tables in Fig 2.

## Data aggregation and pre-processing

To build the basis of the model, we divided Indonesia land mass into rectangle cells, each with size 1-degree × 1-degree (equal to 60-minutes arc). A visualization of sample cell centres is provided in Fig 3. In addition, the temporal resolution of 1-week was considered. This resolution was selected as it offers a good balance between the precision of decision-making and the time required for data processing. The response variable for each cell (i.e. each row of dataset) was classified as zero if there were no AI events within the cell and during the specified

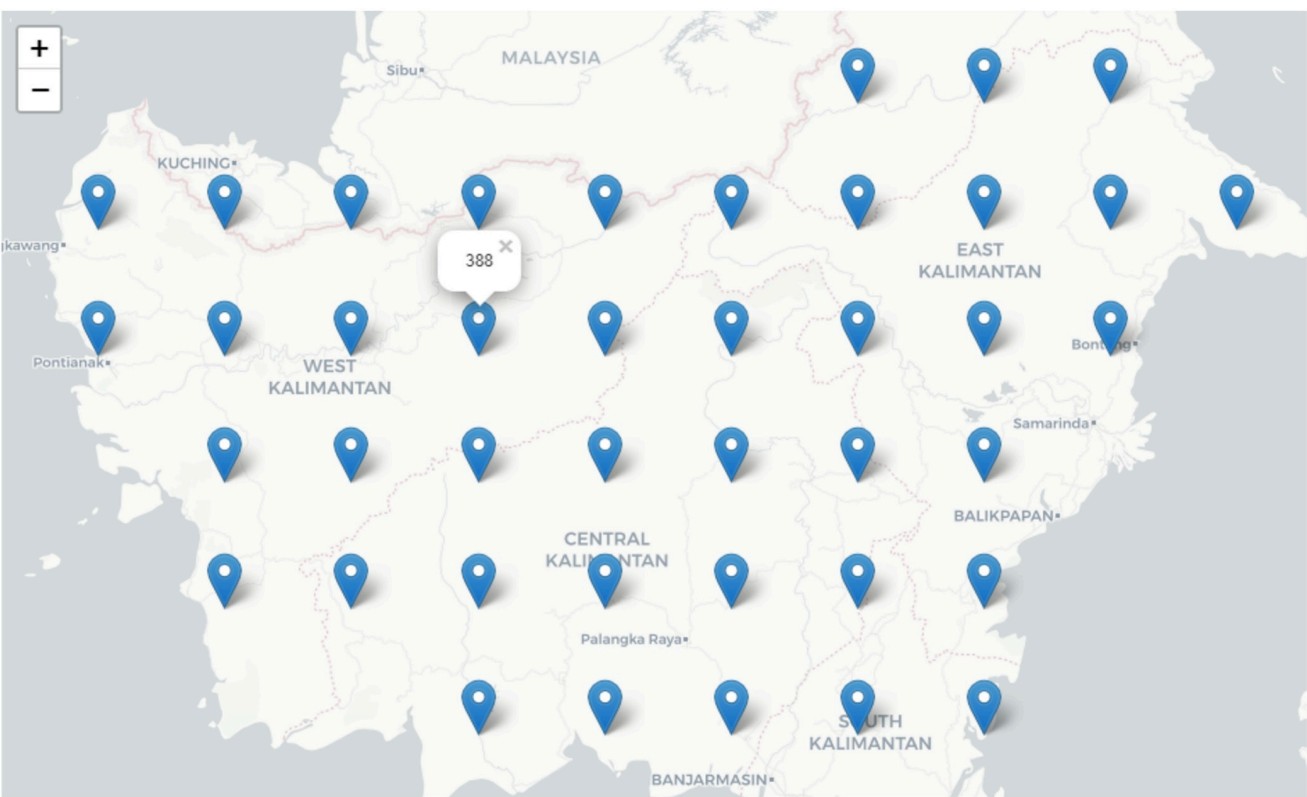

**Fig 3. Sample spatio scale (Indonesia).** Base map and data from OpenStreetMap and OpenStreetMap Foundation.

temporal scale. Conversely, if there was at least one disease event within the spatiotemporal scale, the response variable was classified as one. The response variable is demonstrated by 'has_event' field in the database schema (Fig 2).

The data aggregation was performed in a Structured Query Language table. The response variable describes a disease event that occurred within a certain time interval (week number t) and a spatial compartment with a center of (x,y), where x represents longitude and y represents latitude. The spatiotemporal information in "Indonesia" data tables in Fig 2 are demonstrated by "week", "cell_longitude" and "cell_latitude" attributes. Similarly, for other covariates, pre-processing techniques were applied to transform the records into the defined grid-based table. Since the table included coordinate information for each record, a weekly timeline of information could be visualized in maps using GIS software. Moreover, extracted patterns from the grid-based data could be simple and easy to understand.

The information on AI events in Indonesia was obtained from the Emergency Prevention System for Animal Health (EMPRES) [42, 43]. The reporting date and geographical coordinates of events in the EMPRES-i platform was used to scale the disease events to specified cells.

To make a classifier that could predict the response variable based on predictor variables, we divided the data into training and testing sets. The data from 2009 to 2016 was used for training and the data containing 2006, 2007, 2008 and 2017 was used for testing purposes. Overall, the number of observations in the training dataset was 61,152 (147 [number of cells] × 8 [number of years] × 52 [number of weeks per year]) and in the testing dataset was 30,576.

In total, the training dataset contained 1,860 rows with target variable of one and 59,292 of zero while the testing dataset contained 168 of class one and 30,408 of zero.

In addition, the explanatory variables were obtained for each spatial and temporal dimension. The "Time Machine Request" type of Darksky API was used to retrieve the climatic information. This returns the observed daily weather conditions given a specified date in the past and a location point. The API responses consist of a JSON-formatted object, from which we selected a set of predictors that have been previously known as factors contributing to outbreaks of AI [44–46]. These predictor variables included temperature, precipitation, humidity, wind speed and pressure.

The chicken and duck distribution data for Indonesia were computed using The Gridded Livestock of the World (GLW) [47]. The data offered GeoTIFF format files that were converted to longitude-latitude-value format using the Rasterio library in Python [48] and then imported to a designed database. The spatial resolution of GLW data (a pixel) was higher than the defined spatial resolution of the present study. Thus, the density points inside a cell have been averaged. However, GLW provides low temporal resolution (2010 only) and therefore the densities have been replicated for all data points with a particular spatial resolution.

Birdlife species data included shapefiles that could be visualized by geographical information system (GIS) software such as ArcGIS. We filtered polygons related to 133 duck species. In addition, due to the very large size of the data, we simplified the polygons. This enabled us to decrease the processing time. Finally, in the field called 'bird_existence' in the database, we specified whether each cell was inside a bird polygon or not.

In addition to the aforementioned explanatory variables, we defined winter, spring, summer, fall seasons by dividing weeks into 48-12, 12-24, 24-36 and 36-48, respectively. Indonesia is passed by the equator and the weather can be split into dry (May-September) and rainy (October-April) seasons. In the present study, fall and winter divisions represent the rainy season while spring and summer represent the dry season.

Predictor variables coming from disparate data sources had different spatial and temporal resolutions. Therefore, the variables were arranged with respect to the defined spatial and temporal resolution. When the spatial resolution was higher than a cell or temporal resolution was

higher than a week, we averaged the values. Conversely, when the resolution was lower than a cell or a week, we repeated the same values for all the cells that fit into that resolution. Finally, various data sources were assembled into a database with a uniform spatiotemporal resolution.

## Data analysis

Given the created dataset, we employed RuleFit, Frequent Pattern Growth (FP-Growth) and Prefix-projected Sequential Pattern Mining (PrefixSpan) models to discover hidden rules that might be predictive or indicate dispersion paths of the risk of AI occurrence.

Patterns were extracted in the form of "IF-THEN" rules. The general form of "IF-THEN" rules is demonstrated as follows (Eq 1). Where X is called an antecedent and Y is called a consequent of the rule. The outcome variable (Y) is true if the condition variable (X) is satisfied.

$$IF \ (X \ is \ A) \quad THEN \quad (Y \ is \ B) \tag{1}$$

A rule consists of several interacting risk factors and their ranges. A combination of the extracted rules was used to build the final rule-based classifier.

Given the prepared training set, a supervised ensemble rule learner (RuleFit) was trained to induce rules. RuleFit [49] is a computational algorithm for rule discovery from a large number of candidate risk factors [36]. It generates rules by first exhaustively searching for candidate rules over the potential risk factors in the "rule generation" phase. Rules are generated automatically by traversing each path through a decision tree. Subsequently, the redundant and irrelevant rules are pruned out in the "rule pruning" phase [49, 50].

Among the advantages of the RuleFit algorithm, several points are of note: 1) This algorithm can rank features by their importance. 2) It outputs interpretable rules. 3) RuleFit relies on a non-parametric model, i.e. Gradient Boosting, with fewer modelling assumptions. Moreover, studies comparing rule extraction methods have shown a competitive accuracy of RuleFit [51, 52].

To address the disparity of explanatory variables, these variables were discretized. The categories of very low (VL), low (L), medium (M), high (H) and very high (VH) were calculated based on the histograms of explanatory variables. Moreover, since the dataset was imbalanced, i.e. the number of negative classes was 30 times more than positive classes in the training set, we under-sampled instances of the majority class (one-to-zero ratio of 0.2). Additionally, some data points were discarded due to the high number of missing values. Subsequently, the model was trained with a 5-fold cross-validation. For each subset, the training set was used to learn the rules and the remaining part to evaluate the model.

In each round, we calculated sensitivity, specificity, precision and F-score metrics. Specificity (Eq 3) measures the proportion of actual negatives that have been correctly identified while sensitivity represents the proportion of actual positives that have been accurately identified (Eqs 4 and 2). Precision represents how many selected cases are relevant and the F-score (Eq 5) is a weighted average of the precision and recall. For two-class classifications, there are four possible cases: For a positive class, if the prediction is positive, this is a called true positive (TP) and if negative, it is a false negative (FN). For a negative example, if the prediction is negative, it is called true negative (TN) and if positive, it is a false positive (FP).

Next, we used the unsupervised FP-Growth algorithm for mining the rules from the training set. The algorithm was first proposed by Han *et al*. [53] and it mines the frequent itemsets without candidate generation [54]. The algorithm first compresses the database into a frequent-pattern tree (FP-tree). Then, FP-tree is divided into a set of conditional databases [54]. The FP-Growth algorithm has proven to be time efficient and to consume less memory than the Apriori Algorithm for mining frequent itemset [12, 55].

Extracted rules can be representative of relations between variables in the dataset. The rules indicating a relationship between predictor and response variables were obtained from the FP-Growth algorithm applied to the training set. Similar to the RuleFit, we discretized explanatory variables and additionally, assigned high risk (HR) and low risk (LR) to the response variable for values one and zero, respectively. Following that, the support and confidence criteria were used to select the most important rules. The support of a rule is the number of instances in the dataset that endorses that rule and the confidence indicates the number of times the "IF-THEN" statements are found true.

$$Recall \ or \ Sensitivity = \frac{TP}{(TP + FN)} \tag{2}$$

$$Specificity = \frac{TN}{TN + FP} \tag{3}$$

$$Precision \ or \ Positive \ Predictive \ Value \ (PPV) = \frac{TP}{(TP + FP)} \tag{4}$$

$$F_1 - score = \frac{2 * Precision * Recall}{Precision + Recall} \tag{5}$$

$$F_\beta - score = \frac{(1 + \beta^2) * Precision * Recall}{(\beta^2 * Precision) + Recall} \tag{6}$$

**Mining sequential rules: Event linkage sites.** We performed an additional analysis to understand paths by which disease could be transmitted. Such findings could contribute to gaining a better understanding of the risk of AI occurrence. For this, we prepared transactional datasets for each year from 2010 to 2016. Given a year, the sequence of cells containing events along with ranges of their associated risk factors were calculated for each month of the year. An example of the transactional dataset of 2011 prepared for this analysis is provided in Table 3. For example, in January, a time-line of cells with events starting with cell number 253

**Table 3. A sample of transactional dataset (2011).**

| Month | Sequence of Cells with disease presence |
|---|---|
| Jan | 253: $dC_H, dD_L, t_H, h_H, ws_L, pc_H, pr_L \rightarrow$ |
| | 376: $dC_H, dD_H, t_H, h_M, ws_H, pc_H, pr_L \rightarrow$ |
| | 294: $dC_H, Dk_H, t_H, h_M, ws_H, pc_H, pr_M \rightarrow$ |
| | 355: $dC_H, dD_H, t_H, h_M, ws_H, pc_H, pr_L \rightarrow$ |
| | . . . |
| Feb | 232: $dC_L, dD_L, t_H, h_H, ws_L, pc_M, pr_M \rightarrow$ |
| | 210: $dC_M, dD_L, t_H, h_M, ws_H, pc_M, pr_M \rightarrow$ |
| | 231: $dC_M, dD_L, t_H, h_L, ws_L, pc_M, pr_L \rightarrow$ |
| | 415: $dC_M, dD_H, t_L, h_H, ws_L, pc_M, pr_M \rightarrow$ |
| | . . . |
| Mar | 210: $dC_M, dD_L, t_H, h_H, ws_L, pc_M, pr_M \rightarrow$ |
| | 167: $dC_M, dD_L, t_H, h_H, ws_L, pc_M, pr_M \rightarrow$ |
| | 253: $dC_H, dD_L, t_H, h_H, ws_H, pc_M, pr_M \rightarrow$ |
| | 376: $dC_H, dD_H, t_H, h_M, ws_H, pc_H, pr_M \rightarrow$ |
| | . . . |

is produced. In the Table, symbols of 'dC', 'dD', 't', 'h', 'pc' and 'pr' denote 'density of chickens', 'density of ducks', 'temperature', 'humidity', 'precipitation' and 'pressure', respectively. Moreover, the subscripts 'L', 'M', 'H' denote 'very low or low', 'medium', 'high or very high', respectively.

These datasets were then fed to the PrefixSpan algorithm. PrefixSpan is a well-known sequential pattern mining algorithm [56]. Studies have shown that PrefixSpan, in most cases, outperforms the Apriori-based algorithms such as the GSP (generalized sequential pattern algorithm), FreeSpan (frequent pattern-projected sequential pattern mining), and SPADE (sequential pattern discovery using equivalence classes) [56, 57]. This is because it finds the frequent items after scanning the sequence data for a single time.

The outcome of the analysis, i.e. serial paths of disease spread, can be added to the final knowledge base and contribute to calculating the risk of AI occurrence.

The discovered patterns of this part of the framework were then used in the prediction part as illustrated in Fig 1 to predict the risk of AI presence and evaluate these predictions.

## Prediction

The extracted rules from the RuleFit and FP-Growth algorithms were ordered using their scores and then two groups of rules were combined to be used for defining a classifier. The process of rule extraction and risk prediction is depicted in Fig 4. Due to the highly imbalanced nature of the data set (a ratio of 1:56 of positives to negatives), unseen data points were undersampled with a positive to negative label ratio of 0.2. Subsequently, for each data point, we searched for up to ten first matching rules. Matching rules are those that their antecedent covers the given data point. We used a variable called 'determinant' to determine the class label of data points as shown in Fig 4. Each time, if a matching rule was a risk-increasing type, we added one unit to the defined variable and if it was risk-decreasing type, we subtracted one unit from the variable. Finally, data points with an amount more than an integer threshold ranged from minus four to four were determined as 'at risk of an disease'. Conversely, those with a value less than the threshold were labelled as 'not at risk of disease'. Comparing these labels with actual ones, we evaluated the results with several measures defined in Eqs 2–5.

## Results and discussion

Underlying patterns in form of "IF-THEN" rules along with their respective importance were identified by RuleFit and FP-Growth models. Ordered lists of rules are shown in Tables 4 and 5, respectively.

We used the measures of support, confidence and coefficient to calculate the degree of importance of the rules. Coefficients represent the change in the response variable for one unit of change in the predictor variable. The importance measure for RuleFit is calculated by the multiplication of coefficient and support measures. The rules with positive coefficient are denoted by "increasing" and negative coefficient by "decreasing" risk types as shown in Table 5. The evaluation of outcomes of the RuleFit algorithm with 5-fold cross-validation showed F-score of 64%.

To extract rules using FP-Growth, we separated the dataset based on their target label. For each group, given generated rules from the FP-Growth algorithm, relevant rules were selected using a comparative support criterion, in which we took into account the ratio of instances in high and low risk groups. The rules with the measure greater than a threshold were considered relevant. The extracted rules of the form "risk factors → event occurrence risk" are given in Table 5. Low and high risks are denoted by 'LR' and 'HR' in the table, respectively.

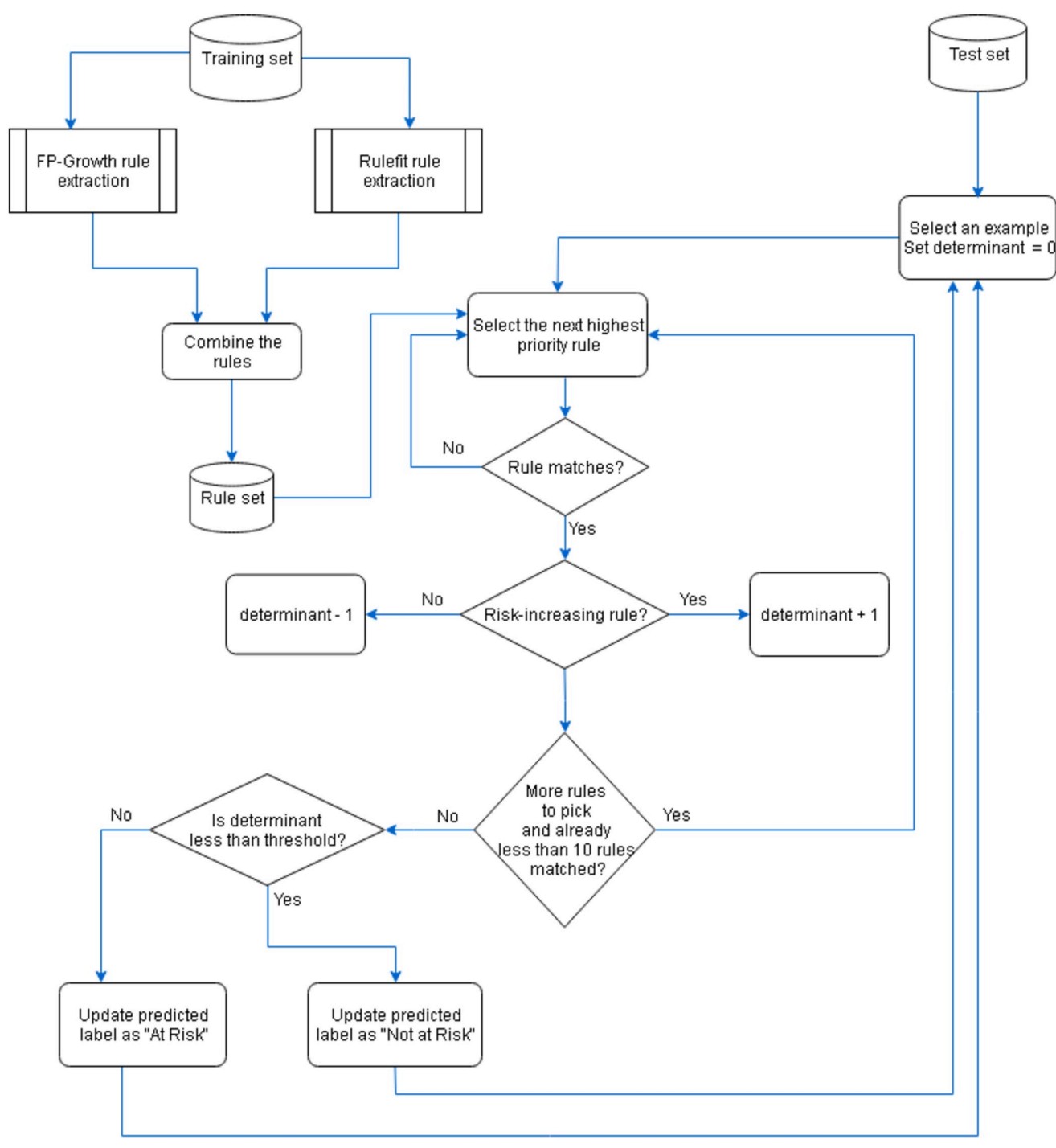

**Fig 4. Rule-based prediction.**

From the most relevant rules obtained from the RuleFit and FP-Growth, we discovered that the FP-Growth and RuleFit algorithms agreed on the impact of several predictors such as chicken density, duck density, season and temperature. Furthermore, these rules were consistent with similar studies [2, 15, 16, 21, 58], which validated our rule-based analysis.

**Table 4. The top risk rules identified by RuleFit.**

| Rule | Effect | Risk Type |
|---|---|---|
| not in the fall season | 2.4045 | decreasing |
| not in the fall season and chicken density not in [H,VH] | 1.5648 | decreasing |
| not in the fall season and duck density not in [H,VH] and temperature is not VH | 0.3915 | decreasing |
| VL duck density | 0.3011 | decreasing |
| duck density not VL and temperature is not VH | 0.2519 | increasing |
| duck density not VL and temperature is not VH and pressure is not in [H,VH] | 0.1819 | increasing |
| not in the fall season and duck density is VL | 0.1680 | decreasing |
| chicken density in [H,VH] and precipitation not in [H,VH] | 0.1371 | increasing |
| duck density not VL and season is not Winter and precipitation not in [H,VH] | 0.1222 | increasing |
| Winter season and chicken density not in [H,VH] | 0.1187 | increasing |
| M chicken density and precipitation is not VH and pressure not in [H,VH] and temperature is not VH | 0.0923 | increasing |

Looking at the extracted rules, it is evident that both algorithms agree on the direct relationship between chicken/duck densities and the risk of AI occurrence. This is consistent with the previous studies [15, 16], in particular the same results have been obtained for developing countries [59] and Indonesia [60].

A low or medium amount of precipitation (less than 300 millimetre per month) was associated with AI occurrence, which was detected by both algorithms. This pattern was aligned with the findings of other studies [12, 19, 20]. Also, both algorithms agreed with regard to finding a connection between rainy season (September-March) and AI occurrence. This might be similar to findings by Loth *et al.* [4] who outlined that wet summers can have a negative association with AI occurrence.

**Table 5. Mined rules by FP-Growth algorithm.**

| Rule | Comparative Support |
|---|---|
| no waterfowls → LR | 824 |
| L chicken density → LR | 634 |
| M pressure and no waterfowls → LR | 555 |
| H temperature and no waterfowl → LR | 537 |
| L chicken density and no waterfowls → LR | 536 |
| L duck density → LR | 519 |
| M precipitation and no waterfowls → LR | 511 |
| VL wind speed and no waterfowls → LR | 467 |
| H chicken density → HR | 433 |
| H temperature and M precipitation → LR | 427 |
| VH humidity → LR | 421 |
| H duck density → HR | 410 |
| VL wind speed and M pressure → HR | 376 |
| Winter season → HR | 367 |
| VH chicken density → HR | 359 |
| M humidity → HR | 335 |
| H chicken density and M pressure → HR | 334 |
| Spring season → HR | 330 |
| H chicken density and H temperature → HR | 324 |

**Table 6. Sample extracted sequential rules by PrefixSpan algorithm.**

| Year | Frequent sequences of cells with disease events |
|------|------------------------------------------------|
| 2010 | $356: dC_H, dD_H, t_H, h_H, ws_L, pc_L, pr_M \rightarrow 253: dC_H, dD_L, t_H, h_H, ws_L, pc_L, pr_M$ |
|      | $253: dC_H, dD_L, t_H, h_H, ws_L, pc_L, pr_M \rightarrow 356: dC_H, dD_H, t_H, h_H, ws_L, pc_L, pr_M$ |
| 2011 | $315: dC_H, dD_H, t_M, h_H, ws_L, pc_H, pr_M \rightarrow 356: dC_H, dD_H, t_H, h_H, ws_L, pc_H, pr_M$ |
|      | $335: dC_H, dD_H, t_H, h_H, ws_L, pc_H, pr_M \rightarrow 315: dC_H, dD_H, t_M, h_H, ws_L, pc_H, pr_M$ |
|      | $356: dC_H, dD_H, t_H, h_H, ws_L, pc_L, pr_M \rightarrow 253: dC_H, dD_L, t_H, h_H, ws_L, pc_L, pr_M$ |
|      | $253: dC_H, dD_L, t_H, h_H, ws_L, pc_L, pr_M \rightarrow 356: dC_H, dD_H, t_H, h_H, ws_L, pc_L, pr_M$ |
| 2012 | $315: dC_H, dD_H, t_M, h_H, ws_L, pc_H, pr_M \rightarrow 356: dC_H, dD_H, t_H, h_H, ws_L, pc_H, pr_M$ |
|      | $335: dC_H, dD_H, t_H, h_H, ws_L, pc_H, pr_M \rightarrow 315: dC_H, dD_H, t_M, h_H, ws_L, pc_H, pr_M$ |
|      | $315: dC_H, dD_H, t_M, h_H, ws_L, pc_M, pr_M \rightarrow 356: dC_H, dD_H, t_H, h_H, ws_L, pc_L, pr_M$ |
|      | $356: dC_H, dD_H, t_H, h_M, ws_M, pc_L, pr_M \rightarrow 315: dC_H, dD_H, t_M, h_H, ws_L, pc_L, pr_M$ |
|      | $356: dC_H, dD_H, t_H, h_H, ws_L, pc_L, pr_M \rightarrow 253: dC_H, dD_L, t_H, h_H, ws_L, pc_L, pr_M$ |
|      | $253: dC_H, dD_L, t_H, h_H, ws_L, pc_L, pr_M \rightarrow 356: dC_H, dD_H, t_H, h_H, ws_L, pc_L, pr_M$ |
|      | $253: dC_H, dD_L, t_H, h_H, ws_L, pc_L, pr_M \rightarrow 355: dC_H, dD_H, t_H, h_M, ws_L, pc_L, pr_M$ |

While the RuleFit algorithm was able to detect the negative association between temperature and AI presence that has been previously outlined in the literature [5, 58], this association was not found using the FP-Growth algorithm.

In an additional analysis, we explored frequent sequences of cells with disease occurrence. A sample of outcome results for 2010 to 2012 is given in Table 6 and also visualized in Fig 5. Outcomes indicate the path between the regions of Lampung, West Java, Central Java, Yogyakarta and East Java. The most frequent paths were between the Lampung and Yogyakarta, West Java and Yogyakarta, East Java to Central Java, Lampung to Central Java, Central Java to West Java and West Java to Lampung.

## Risk prediction

The classifier was parametrized with a range of thresholds as explained earlier and a precision-recall curve was generated. The curve is visualized in Fig 6 and represents the trade-off

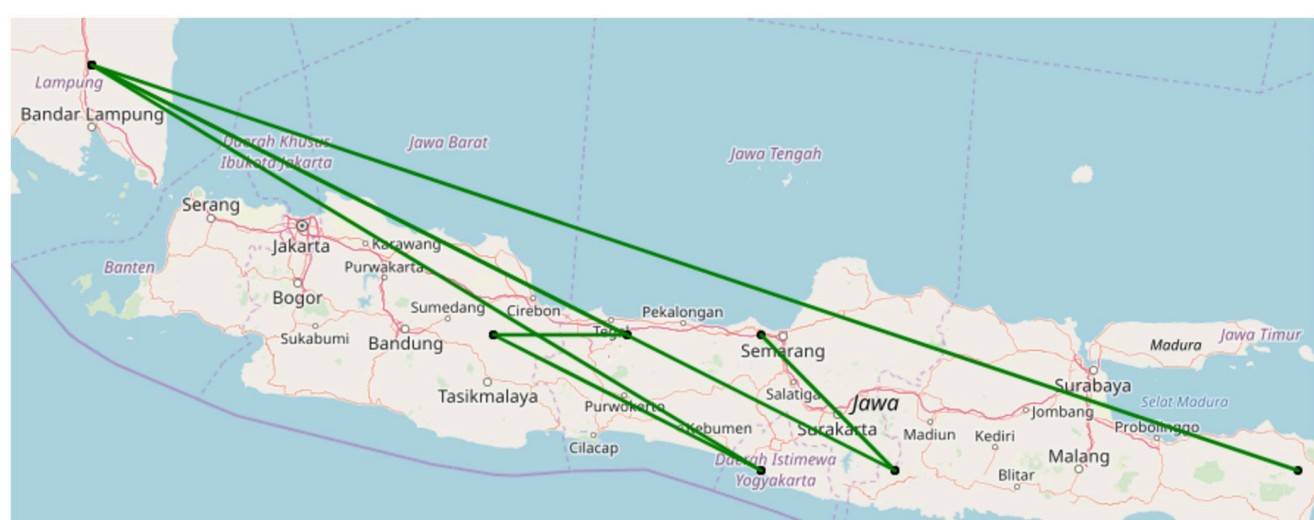

**Fig 5. Linkage sites of disease events.**

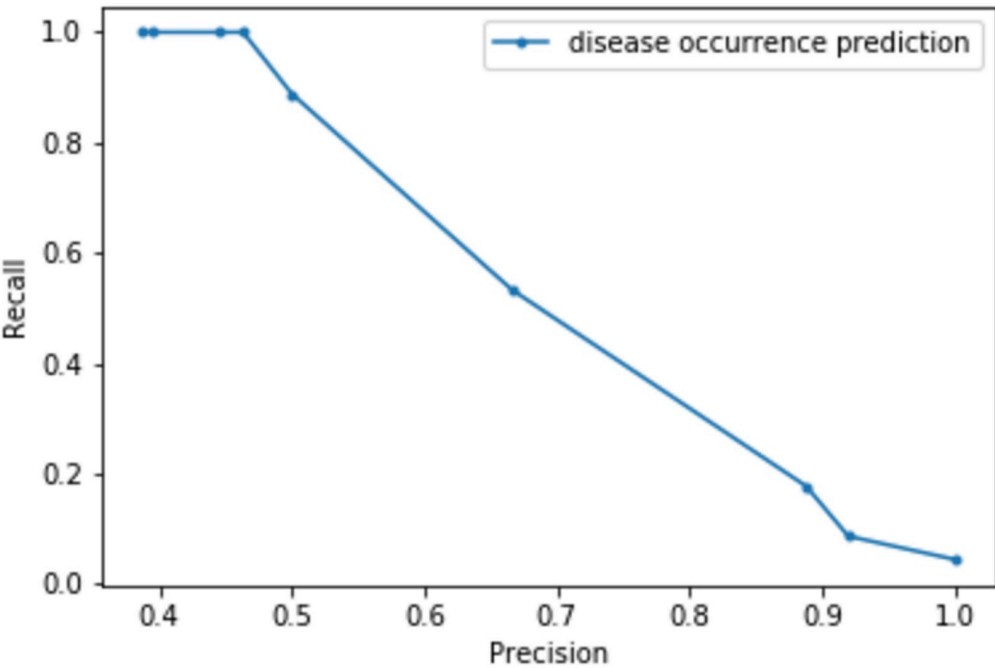

**Fig 6. Precision-recall curve.**

between precision and recall measures of predictions. The blue dot points show the thresholds starting from minus four at the left and ending to four at the right of the graph. As the threshold is increased, the sensitivity decreases and the precision increases.

Since the goal of the classifier defined here is to predict disease event occurrence, having a high sensitivity is important. This is because, in such prediction systems, having less false negatives is more desirable than having less false positives. Therefore, a threshold of zero was selected.

The graph visualized in Fig 6 represents the trade-off between precision and recall measures. Based on Fig 6, at the threshold of zero, the classifier gains a sensitivity of about 88% and a positive predictive value of 50%. The high sensitivity means that the classifier is strong for correctly predicting the disease presence. The system also predicts actual disease absent points with a probability of 82.22%. These results are summarized in the Table 7.

Since the dataset here contains more negative than positive classes, precision and recall are better metrics to look at. A refined version of $F_1$-score called $F_\beta$-score (Eq 6) is more practical for imbalanced data since it allows for higher weighting of either precision or recall. Given the importance of sensitivity over precision in the current study, in addition to traditional $F_1$-score, $F_\beta$-score with $\beta = 2$ was reported.

**Table 7. Evaluation measures of the rule-based classifier (threshold = 0).**

| Measure | Value |
|---|---|
| Sensitivity | 88.88% |
| Specificity | 82.22% |
| Positive predictive value | 50% |
| $F_1$-score | 64% |
| $F_\beta$-score | 76.92% |

**Table 8. Evaluation measures of the a Random Forest classifier (number of trees = 20).**

| Measure | Value |
|---|---|
| Sensitivity | 56.8% |
| Specificity | 82.22% |
| Positive predictive value | 80% |
| $F_1$-score | 63.3% |
| $F_\beta$-score | 58.9% |

To compare the performance results with a basic classifier, we applied the Random Forest algorithm with 10-fold cross-validation on unseen data. The result reported in Table 8 shows that the performance of the proposed classifier is comparable with Random Forest. Although both classifiers gained the same F-measure, giving more importance to sensitivity (i.e. β = 2), the proposed model obtained a higher $F_\beta$-score. It should be noted that different resolutions and case studies in previous studies impede us to make a direct comparison of performance with them [61].

Moreover, we generated the cumulative gains curve, which was used to assess the performance of the prediction. It shows the percentage of targets reached when considering a certain percentage of the population. First, all the observations were ordered according to the output of the model. Therefore, observations with the highest rank were placed on the left-hand side of the horizontal axis. The vertical axis of the curve indicates which percentage of true positives included in the curve.

The chart can tell how much the model predicts better compared to a random selection. According to Fig 7, if we consider the 20% of the observations, the model will ensure that 88.89% of the true positives are in this group, while the random pick would provide only the 20% of the targets.

## Conclusion and future work

Here, we have proposed a framework to discover hidden patterns from an extensive list of data sources using rule-discovery techniques. This framework facilitates the understanding of how AI predictors and occurrence data can be aggregated and pre-processed as input for rule-discovery techniques. Subsequently, a classifier was built from extracted rules to predict the disease presence in new circumstances. This approach is complementary to existing AI risk profiling methods. A rule-set used here can offer easier interpretations of predictions for end-users. This means that users can understand how predictions are made. Also, it is easy to identify which factors have contributed to the predictions and whether the predictions are reasonable. An understanding of risk-increasing factors and their interactions and building risk-profiling maps, can be useful in emergency preparedness. For instance, authorities may use such information to prioritize and target areas for interventions.

The outcome patterns in this study were consistent with earlier studies. For example, the positive association between disease presence and chicken/duck densities, waterfowl density and the negative association between disease presence and precipitation were aligned with previous studies. Nevertheless, the impact of temperature on disease occurrence showed contradictory results, which might be due to the tropical climate of Indonesia. In Indonesia, the temperature is usually high and does not change much during a year.

An important limitation of the present study is that the change in the distribution of data through the process of under-sampling without considering the impact of imbalanced data on the classification output might be misleading [62]. This is because the removal of examples

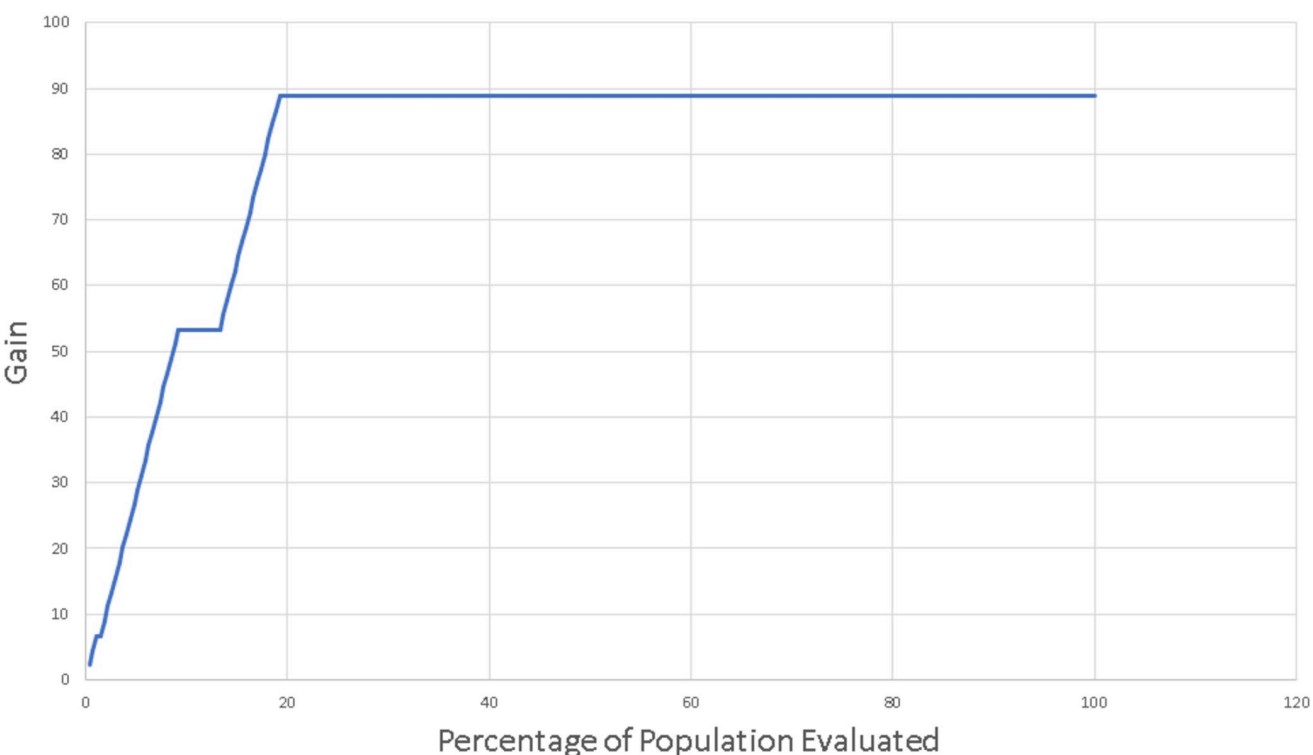

**Fig 7. Cumulative gains curve.**

from the majority class could lead to the loss of potentially important information about the class [63]. Moreover, further work is required to improve the performance of predictions. One approach is the collection of additional data sources. For example, live bird trades information could be taken into consideration. Trading of live birds is known to be a major pathway of AI transmission that can happen through the movements of contaminated traders [64]. To improve the timeliness of predictions, a continuous pipeline from data collection to analysis is required. It means that during specified time intervals, data is automatically collected, pre-processed, integrated and analysed. The patterns in the rule-base will be updated in each interval, which ensures real-time predictions.

The proposed framework may provide public health officials and animal health authorities with warnings that can be used for identifying areas with a high risk of disease presence. Such information can potentially be used for response in high priority areas and executing interventions.

## Supporting information

**S1 File.**
(ZIP)

## Author Contributions

**Conceptualization:** Rozita Dara, Shayan Sharif.

**Data curation:** Samira Yousefinaghani.

**Formal analysis:** Rozita Dara.

**Methodology:** Samira Yousefinaghani, Rozita Dara, Shayan Sharif.

**Supervision:** Rozita Dara, Shayan Sharif.

**Validation:** Samira Yousefinaghani, Rozita Dara, Zvonimir Poljak, Fei Song.

**Writing – original draft:** Samira Yousefinaghani.

**Writing – review & editing:** Rozita Dara, Zvonimir Poljak, Fei Song, Shayan Sharif.

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
