## [Decision Letter · Decision Letter 0]

9 Oct 2020

PONE-D-20-03935

A Framework for Risk Assessment of Avian In Occurrence: An Indonesian Case Study

PLOS ONE

Dear Dr. Dara,

Thank you for submitting your manuscript to PLOS ONE. After careful consideration, we feel that it has merit but does not fully meet PLOS ONE’s publication criteria as it currently stands. Therefore, we invite you to submit a revised version of the manuscript that addresses the points raised during the review process.

Both reviewers raise concerns about the submission. The first reviewer, who recommend rejection, is dubious about the actual contribution of the work, which should be properly highlighted and discussed. The second reviewer, on the other hand, recommend that the paper is thoroughly revised, especially to frame it in the correct conceptual context. I think the two reviews are not in contrast, and rather complement each other. I invite the authors to proceed with an extensive and thorough review of the manuscript, which will be send back for a further review to the original reviewers, if possible. 

We look forward to receiving your revised manuscript.

Kind regards,

Alessandro Rizzo

Academic Editor

PLOS ONE

Journal Requirements:

2.We note that [Figure(s) 2 and 3] in your submission contain [map/satellite] images which may be copyrighted. All PLOS content is published under the Creative Commons Attribution License (CC BY 4.0), which means that the manuscript, images, and Supporting Information files will be freely available online, and any third party is permitted to access, download, copy, distribute, and use these materials in any way, even commercially, with proper attribution. For these reasons, we cannot publish previously copyrighted maps or satellite images created using proprietary data, such as Google software (Google Maps, Street View, and Earth). For more information, see our copyright guidelines: http://journals.plos.org/plosone/s/licenses-and-copyright.

1.    You may seek permission from the original copyright holder of Figure(s) [2 and 3] to publish the content specifically under the CC BY 4.0 license. 

4.Thank you for stating the following in your Competing Interests section: 

[No].

5. .We note that you have indicated that data from this study are available upon request. PLOS only allows data to be available upon request if there are legal or ethical restrictions on sharing data publicly. For information on unacceptable data access restrictions, please see http://journals.plos.org/plosone/s/data-availability#loc-unacceptable-data-access-restrictions.

Additional Editor Comments (if provided):

Both reviewers raise concerns about the actual contribution of the work and the proper context in which the submission would fit. I invite the authors to proceed to a thorough revision of their work and resubmit it.

Reviewers' comments:

Reviewer's Responses to Questions

**Comments to the Author**

1. Is the manuscript technically sound, and do the data support the conclusions?

Reviewer #1: Yes

Reviewer #2: Partly

2. Has the statistical analysis been performed appropriately and rigorously? 

Reviewer #1: No

Reviewer #2: N/A

3. Have the authors made all data underlying the findings in their manuscript fully available?

Reviewer #1: No

Reviewer #2: No

4. Is the manuscript presented in an intelligible fashion and written in standard English?

Reviewer #1: Yes

Reviewer #2: Yes

5. Review Comments to the Author

Reviewer #1: The paper treats the problem of training data set to predict virus diffusion and influence with respect to a time windows of several years. The application proposal is reported about data of a secific case, Indonesian data. A framework for data acquisition composed mainly of three modules is reported. Data acquisition, data management (relational db) and data processing and prediction.

The database is a very simplified relational db where geographic data are managed as numbers. This is a risky simplification for geographic data. But beside this, the proposed procedure - to the best of the reviewer's knowledge - is a standard analysis and there is little contribution with respect to prediction tools. Geographic information are used and proposed as guide for managing influenza diffusion.

Case analysis is reported in a too simple if then else structure. Dataset and features selected are not exaclty clear both in the way they are identified as well as the description. Many parts refer to existing tools so it is difficult to gather the right contribution. Also dataset are not available as well as tool. Latter may be useful for a similar application.

With the pandemia problem, how data and information may be related to virus diffusion may be relevant.

Reviewer #2: Thank you for inviting me to review this manuscript describing rule based methodology and its application to avian influenza outbreaks in Indonesia using data on climatic variables, bird species distribution and poultry density. Overall, I think it is an interesting description of the model process but feel that it would benefit the paper if the application of the model itself is promoted rather than its usefulness as a prediction for avian influenza outbreaks.

Major issues:

As it stands the paper does not describe ‘risk assessment’ in its traditional sense rather risk factors or risk prediction. For this reason I would suggest changing the title to something like ‘Application of a rule based prediction model for avian influenza outbreaks in Indonesia’. It might be good to include in the future work section an example of where the model could be applied in a situation where the link between risk factors and disease are more evident. For avian influenza this is quite difficult. You quote Loth et al., 2010 who found that “roads” was a significant risk factor. However, looking more in depth he states that it is ‘Possible contaminated transport vehicles return dirty egg-trays back to the farms, increasing the chance of disease spread. Increased movement over longer distances may increase the disease spread even further. This may explain why “roads” was a significant risk factor.’ By using prediction models such as this one the true picture of the relationship between the risk factors can be missed. For this reason I disagree with the statement that it provides a more holistic understanding of the drivers of AI epidemics. It might also be useful to add a paragraph on how to mitigate outbreaks once they have been predicted e.g. if you know an area has high poultry density and the model predicts high risk, how can this be addressed? Would a campaign for better biosecurity in this area help?

Minor issues:

Overall:

• Throughout the paper please abbreviate avian influenza (AI) when first mentioned and then use abbreviation throughout.

• Please be consistent with temporal/time or spatial/space

• How does it take account that there might be a third variable involved e.g. precipitation=standing water=virus survival? See comment above about relationship between risk factors

Abstract:

• Suggest change ‘had’ to ‘have’

• Suggest change ‘to be transmitted to humans’ to ‘for zoonotic transmission’

• Have disease prediction systems actually been used in reality to mitigate the consequences of avian influenza?

• ‘In this study, we have proposed a framework for the prediction of the occurrence and spread of avian influenza events in a geographical area.’ This doesn’t match up with the title. Please see comment above

• ‘Comprehensive list’ implies you used more than 4 data sources and 8 factors. I’m not sure it is comprehensive

Introduction:

• Pg 3 Have there been global efforts to eradicate AI? Don’t we just vaccinate humans against influenza annually, so this is control only?

• Pg 3 Suggest change ‘Had’ threatened human lives’ to ‘has’

• Pg 3 ‘To mitigate the impact of avian influenza outbreaks, it is necessary to understand the extent to which different risk factors and their interactions contribute to the introduction and spread of outbreaks.’ Overall I’m not sure how any of the sources have in reality been used to mitigate the impact of RA outbreaks? I think the summary here of the contradiction between studies in the association with risk factors proves how difficult it is or proves that there is perhaps a more complex interaction of factors at play.

• Pg 3 ‘The impact of environmental factors and climate change on the spread and geographical distribution of avian influenza outbreaks is well-known in the literature.’ Not sure this is impact more predictive value

• Pg 3 Suggest change ‘well known’ to ‘Documented’

• Pg 3 Suggest change ‘contrary’ to ‘contrast’

• Pg 3 ‘is positively connected with H5N1’ suggest insert’ HPAI strain’ before H5N1

• Pg 4 ‘However, Pavade et al. (2011) found connected poultry density to avian influenza outbreaks only for least developed countries that do not have economic growth and financial stability.’ Delete ‘found’. Not sure this is entirely true: is it not more connected to smallholder type farming and poorer between and within farm biosecurity?

Motivation:

• Pg 5 ‘Clearly, the system only provides historical information and the temporal resolution is not high enough.’ Not high enough for what?

• Pg 5 The proposed framework was implemented and tested for an Indonesian case study. Indonesia was selected as a case study as this country has had a high number of avian influenza outbreaks over the years and, importantly, it provides accessible explanatory data sources. Delete full stop and words in red.

• Table 1: possible to put website links in?

• Table 2: atmospheric pressure?

Data Aggregation and Pre-processing

• Pg 8 curious to know why data from 2009 to 2016 was used for training and 2006, 2007, 2008 and 2017 was used for testing?

• Pg 9 (?) should this be (GLW)?

• Pg 9 why only duck species?

• Pg 9 If the model already accounts for temperature, precipitation, humidity etc. why also take account of seasons – wont these implicitly be accounted for using the climatic variables?

• Pg 9 – were any scenario analyses carried out to look at the effect of averaging or repeating the values as described?

Data Analysis:

• Pg 11 ‘this is a called a true positive’ delete ‘a’

Prediction:

• Pg 13 Suggest change ‘Due to high imbalanced nature of the data set’ to ‘Due to the highly imbalanced nature’

Results and discussion:

• Pg 16 delete ‘as well’ after Belkhiria et al., 2018)

Conclusion and future work:

• ‘comprehensive list’ – same comment as before

• In the methodology section it states: ‘In the last step (Prediction), end-users can communicate with the system through a user interface. Here, the user interface can include mapping and monitoring of the risk, given a current spatio-temporal dataset.’ Where is the link to the user interface? Have any maps been generated from this work?

References:

• Please check references for H and N rather than h and n.

6. PLOS authors have the option to publish the peer review history of their article (what does this mean?). If published, this will include your full peer review and any attached files.

Reviewer #1: No

Reviewer #2: No

---

## [Author Response · Author response to Decision Letter 0]

13 Nov 2020

Date: 02 Nov 2020

Title: A Framework for Risk Assessment of Avian Influenza Occurrence: An Indonesian Case Study

Authors: Samira YousefiNaghani, Rozita Dara, Zvonimir Poljak, Fei Song, and Shayan Sharif

Manuscript No: PONE-D-20-03935

Dear Dr. Alessandro Rizzo

First, we would like to thank you, Academic Editor, and the reviewers for the constructive review and comments forwarded to us with regard to the Manuscript ID PONE-D-20-03935 entitled "A Framework for Risk Assessment of Avian Influenza Occurrence: An Indonesian Case Study". 

We have addressed the comments and made the required changes which we believe have improved the manuscript in a way that is now acceptable for publication in PLOS ONE journal. In the following, we will describe our responses to the points made by academic editor and reviewers. Given comments have been copied and are followed by specific responses in green. 

Academic Editor comments:

Authors:

• The figure naming has been changed. We replaced “Figure” with “Fig” and “Equation” with “Eq” to match the formatting.

• We made the Figures titles bold and each comes after the paragraph they were mentioned for first time. In addition, titles were added in those sections that figures should be located. 

• Double-space between paragraphs was set.

• Tables were moved immediately after the paragraph they were cited for first time and titles bolded.

• Citations changed to number format and accordingly the name of first author et al. was added where necessary.

• Tables modified to cell-based format

• Author contribution section added.

• Numbers were removed from section headings.

2.We note that [Figure(s) 2 and 3] in your submission contain [map/satellite] images which may be copyrighted. All PLOS content is published under the Creative Commons Attribution License (CC BY 4.0), which means that the manuscript, images, and Supporting Information files will be freely available online, and any third party is permitted to access, download, copy, distribute, and use these materials in any way, even commercially, with proper attribution. For these reasons, we cannot publish previously copyrighted maps or satellite images created using proprietary data, such as Google software (Google Maps, Street View, and Earth). 

Authors: The Figures 2 and 3 were created by us and were not previously copyrighted anywhere. We created the database schema (Figure 2) with MySQL Workbench 8. And the map (Figure 3) with Python Folium library. 

Authors: We used the PLOS LATEX template for the current submission.

4.Thank you for stating the following in your Competing Interests section: 

[No].

Authors: We have addressed this issue.

Authors: We have addressed this issue.

Authors: We have addressed this issue.

Authors: A list of supplementary tables is given in “supporting Information” and cited in the sentence at the end of first paragraph in section “Data Collection”.

Additional Editor Comments (if provided):

Both reviewers raise concerns about the actual contribution of the work and the proper context in which the submission would fit. I invite the authors to proceed to a thorough revision of their work and resubmit it.

Authors: The contribution of this work is explained in the Motivation in the Introduction Section. The main contributions include: 

(1) The study uses several data sources with high temporal and spatial resolutions that can enhance the accuracy of results. (2) It employs rule-discovery models rather than focusing on predictive regression models commonly used in the relevant existing studies. Rule-discovery models generate patterns that can link the risk of disease presence with subsets of risk factors. (3) It contributes to automation and transparency of predictions. To make it clearer we added the lines below at the end of the second last paragraph in Motivation. Although transparency of predictions is essential in application areas of epidemiology, the transparency of surveillance systems and their outcomes has received less attention in the literature. Since the proposed framework used a collection of rules as a high-level description of data, the explanatory capability of predictions can be enhanced. This means public health officials can find the reasons behind the predictions made by the system.

 (4) The database is a combination of disparate data sources with different scales. We performed a complex transformation on data sources to make them scalable in the defined cells.

Reviewer #1: The paper treats the problem of training data set to predict virus diffusion and influence with respect to a time windows of several years. The application proposal is reported about data of a secific case, Indonesian data. A framework for data acquisition composed mainly of three modules is reported. Data acquisition, data management (relational db) and data processing and prediction.

The database is a very simplified relational db where geographic data are managed as numbers. This is a risky simplification for geographic data.

Authors: Thanks for your helpful comments! 

The database we used is a combination of disparate data sources with different temporal and spatial scales. To the best of our knowledge, previous studies have used single sources of data , combined various analysis methods not data sources, or found the impact of each risk factor on the output separately . We performed a complex transformation on data sources to make them scalable in the defined cells (with specific temporal and spatial resolution). Data transformation has been achieved with extensive feedback from the subject matter experts such as epidemiologists and immunologists. For example, the birdlife-species data consisted a heavy collection of polygons that we simplified and then found out if each cell fall in any polygon area. We selected a spatial resolution of (1-degree) * (1-degree) and a temporal resolution of 1-week to balance between the computational expenses and precision of decision-making. We have examined various methods for transforming this data source and believe that there would be limited variability in geographical data within the specified time and space. 

 But beside this, the proposed procedure - to the best of the reviewer's knowledge - is a standard analysis and there is little contribution with respect to prediction tools. Geographic information are used and proposed as guide for managing influenza diffusion.

Authors: To the best of our knowledge, our study is the first to use rule-based methods (Rulefit, FP-Growth and Prefix-span) to extract knowledge from an extensive combination of risk factors for prediction of avian influenza outbreaks. Many spatiotemporal prediction studies are based on regression models and do not consider the interaction between risk factors. Rule-based methods benefit the predictions in two ways: 1) the interactions between risk factors were taken into account. 2) they add transparency to predictions. We believe that transparency of predictions is essential in application areas of epidemiology. Our proposed framework used a collection of rules as a high-level description of data which enhances the explanation capability of predictions. This will be beneficial because public health officials will be able to find the reasons behind the predictions made by the system.

Case analysis is reported in a too simple if then else structure. Dataset and features selected are not exaclty clear both in the way they are identified as well as the description. Many parts refer to existing tools so it is difficult to gather the right contribution. Also dataset are not available as well as tool. Latter may be useful for a similar application.

Authors: A complete description of features for each table used in our database is given in the supplementary files. To select data sources and tools for this study, we reviewed the literature and identified data attributes and software tools which best suited our research. We also involved subject matter experts and end-users (e.g. epidemiologists) in different phases of study and attempted trial and errors to select the tools, extract data attributes, and select model/algorithms and their parameters. 

Authors: We will upload the aggregated data with this revision of the manuscript. 

With the pandemia problem, how data and information may be related to virus diffusion may be relevant.

Authors: In addition to factors such as personal hygiene and airlines, coronavirus can change its transmission behaviour by environmental factors such as humidity, temperature and wind speed (see papers and ). Therefore, the fundamental of this study can be used for the current situation.

Reviewer #2: Thank you for inviting me to review this manuscript describing rule based methodology and its application to avian influenza outbreaks in Indonesia using data on climatic variables, bird species distribution and poultry density. Overall, I think it is an interesting description of the model process but feel that it would benefit the paper if the application of the model itself is promoted rather than its usefulness as a prediction for avian influenza outbreaks.

Major issues:

As it stands the paper does not describe ‘risk assessment’ in its traditional sense rather risk factors or risk prediction. For this reason I would suggest changing the title to something like ‘Application of a rule based prediction model for avian influenza outbreaks in Indonesia’. It might be good to include in the future work section an example of where the model could be applied in a situation where the link between risk factors and disease are more evident. For avian influenza this is quite difficult. You quote Loth et al., 2010 who found that “roads” was a significant risk factor. However, looking more in depth he states that it is ‘Possible contaminated transport vehicles return dirty egg-trays back to the farms, increasing the chance of disease spread. Increased movement over longer distances may increase the disease spread even further. This may explain why “roads” was a significant risk factor.’ By using prediction models such as this one the true picture of the relationship between the risk factors can be missed. For this reason I disagree with the statement that it provides a more holistic understanding of the drivers of AI epidemics. It might also be useful to add a paragraph on how to mitigate outbreaks once they have been predicted e.g. if you know an area has high poultry density and the model predicts high risk, how can this be addressed? Would a campaign for better biosecurity in this area help?

Authors: Thanks for your helpful suggestions! Below find the modifications we made. 

• To address your comment about risk assessment, we changed the title to ‘A Framework for the Risk Prediction of Avian Influenza Occurrence: An Indonesian Case Study’. 

• To address your concern on using of words such as ‘holistic’ or ‘comprehensive’, we replaced them with words such as ‘broad’ or ‘extensive’ throughout the text. 

• We also added the following lines in the conclusion to give an example on how predictions can be used to mitigate outbreaks.

“An understanding of risk-increasing factors and their interactions and building risk-profiling maps, can be useful in emergency preparedness. For instance, authorities can use such information to prioritize and target areas for interventions.”

Minor issues:

Overall:

• Throughout the paper please abbreviate avian influenza (AI) when first mentioned and then use abbreviation throughout.

Authors: We have now used AI instead of Avian influenza after the first time and highlighted throughout the text.

• Please be consistent with temporal/time or spatial/space

Authors: We replaced time and space with temporal and spatial and highlighted the changes in the text.

• How does it take account that there might be a third variable involved e.g. precipitation=standing water=virus survival? See comment above about relationship between risk factors

Authors: Thanks for this great point. We acknowledge that there may be other variables involved in contributing to the risk factor. However, out study is a proof of concept in which we didn’t consider all possible variables. Furthermore, data availability was a challenge that we had to deal with for this research. In fact, ssome of these data sources that we were hoping to collect for our research were not even accessible. 

Abstract:

• Suggest change ‘had’ to ‘have’ 

Authors: We change ‘had’ to ‘have’ and highlighted in the text. 

• Suggest change ‘to be transmitted to humans’ to ‘for zoonotic transmission’

Authors: We changed it as per your recommendation. 

• Have disease prediction systems actually been used in reality to mitigate the consequences of avian influenza?

Authors: Prediction systems have been used to prioritize the control actions for high risk areas. For example, if bird trade as a high impact as the risk factor of avian influenza, administrators may decide to restrict trades in order to mitigate the spread of the virus. Another example can be about poultry density. if poultry density has a positive association with avian influenza outbreaks, the premises with a high density of poultry would be more at risk and might be vaccinated, for example.

We added a paragraph in the ‘Conclusions’ to describe how the predictions can help in mitigating the consequences.

• ‘In this study, we have proposed a framework for the prediction of the occurrence and spread of avian influenza events in a geographical area.’ This doesn’t match up with the title. 

Authors: In fact, we first performed a risk assessment and the then predicted the risk of avian influenza occurrence. To address your comment, we have changed the title to ‘A Framework for the Risk Prediction of Avian Influenza Occurrence: An Indonesian Case Study’.

• ‘Comprehensive list’ implies you used more than 4 data sources and 8 factors. I’m not sure it is comprehensive

Authors: We changed it to ‘extensive’ in the abstract, the first paragraph of Motivations and the first paragraph of Conclusions. We also changed the ‘a more holistic understanding’ in the second paragraph of the abstract to ‘a broad understanding’.

Introduction:

• Pg 3 Have there been global efforts to eradicate AI? Don’t we just vaccinate humans against influenza annually, so this is control only?

Authors: Avian influenza is usually controlled by mass vaccination, culling or moving restrictions. Here, by ‘global efforts’ we meant the research works that have been conducted to understand the virus transmission mechanism and seek a solution for its eradication. We changed the global to ‘researchers’ In the text.

• Pg 3 Suggest change “Had” threatened human lives’ to “has”

Authors: We changed ‘had’ to ‘has’ and highlighted in the text.

• Pg 3 ‘To mitigate the impact of avian influenza outbreaks, it is necessary to understand the extent to which different risk factors and their interactions contribute to the introduction and spread of outbreaks.’ Overall I’m not sure how any of the sources have in reality been used to mitigate the impact of RA outbreaks? I think the summary here of the contradiction between studies in the association with risk factors proves how difficult it is or proves that there is perhaps a more complex interaction of factors at play.

Authors: 

We agree with your statement about inconsistencies of findings in the literature on the association of risk factors. Such inconsistencies may suggest a more complex interaction between the risk factors. We made an attempt to avoid this in our research by utilizing rule-based methods to identify these interactions better. 

In general, the impact of risk factors is used to identify the possible outbreak clusters so that this information can help prioritize and target the areas that control strategies such as vaccination need to be applied.

• Pg 3 ‘The impact of environmental factors and climate change on the spread and geographical distribution of avian influenza outbreaks has been shown in the literature.’ Not sure this is impact more predictive value

Authors: We added some references at the end of the sentence ‘The impact of environmental factors and climate change on the spread and geographical distribution of avian influenza outbreaks has been shown in the literature.’ , which have indicated that environmental factors contribute to the spread of the avian influenza. 

• Pg 3 Suggest change ‘well known’ to ‘Documented’

Authors: Thanks for the suggestion. ‘Well-known’ is changed to ‘documented’ and highlighted in the text.

• Pg 3 Suggest change ‘contrary’ to ‘contrast’

Authors: Thanks for bringing this up. This change has been made. 

• Pg 3 ‘is positively connected with H5N1’ suggest insert’ HPAI strain’ before H5N1

Authors: We added HPAI before H5N1 and highlighted this change in the text.

• Pg 4 ‘However, Pavade et al. (2011) found connected poultry density to avian influenza outbreaks only for least developed countries that do not have economic growth and financial stability.’ Delete ‘found’. Not sure this is entirely true: is it not more connected to smallholder type farming and poorer between and within farm biosecurity?

Authors: We removed the sentence from the paper.

Motivation:

• Pg 5 ‘Clearly, the system only provides historical information and the temporal resolution is not high enough.’ Not high enough for what?

Authors: To clarify, ‘the temporal resolution is not high enough’ was changed to ‘the monthly average of climatic variables is a low temporal resolution’

• Pg 5 The proposed framework was implemented and tested for an Indonesian case study. Indonesia was selected as a case study as this country has had a high number of avian influenza outbreaks over the years and, importantly, it provides accessible explanatory data sources. Delete full stop and words in red. 

Authors: Unfortunately, we were not able to address this comment as we couldn’t find the text that was highlighted in red. 

• Table 1: possible to put website links in?

Authors: References 39 to 41 are now cited in Table1.

• Table 2: atmospheric pressure? 

Authors: We used the sea-level air pressure in mbar. Each atm is equal to 1013.25 mbar. 

For more information visit: https://en.wikipedia.org/wiki/Atmospheric_pressure

Data Aggregation and Pre-processing

• Pg 8 curious to know why data from 2009 to 2016 was used for training and 2006, 2007, 2008 and 2017 was used for testing?

Authors: There was no difference in the number of instances or distribution of values among years. Initially, we used 2006-2008 for testing and 2009-2016 for training. Later, since we aimed to use the result of test data for a decision support system that combined social media rules with spatiotemporal rules for 2017, we built the spatiotemporal dataset for 2017 as well. 

• Pg 9 (?) should this be (GLW)? 

Authors: That is correct. We changed and highlighted it.

• Pg 9 why only duck species? 

Authors: Duck species were selected in this study as they are an important group in epidemiology and ecology of avian influenza, they are abundant, and they are migratory. In addition, to reduce variability in the datasets and consistency of our analysis, we considered only duck species as migratory birds.

• Pg 9 If the model already accounts for temperature, precipitation, humidity etc. why also take account of seasons – wont these implicitly be accounted for using the climatic variables?

Authors: We think that seasons in Indonesia are not predictable by climatic variables possibly due to global warming. Moreover, usability of the end product (i.e. decision support systems) was another factor to take seasons into account. In other words, seasons would be a more understandable factor for the end-users of a decision support framework.

• Pg 9 – were any scenario analyses carried out to look at the effect of averaging or repeating the values as described?

Authors: In this study, in order to aggregate different sources prior to analyses, we divided the Indonesia area to cells with 1*1-degree spatial and 1-week temporal resolution. In order to assign a specific value of each risk factor in each cell, we used averaging and repeating. Basically, if we had multiple values inside a cell (higher resolution than a cell), we averaged the values to come to a single value. In fact, we assume that all points in a cell have similar properties. Conversely, if we had a single value for several cells (lower resolution than a cell), we repeated that value for all cells in that low resolution. Here, we assume that all cells within that resolution have the same properties.

Data Aggregation:

Authors: We made an additional change: The sentence ‘This resolution was selected as it offers a good balance between the precision of decision-making and the time required for data processing.’ has been moved from the end of the second paragraph to the first paragraph of ‘Data Aggregation’, which we thought would be a better place for the sentence. We highlighted the change.

Data Analysis:

• Pg 11 ‘this is a called a true positive’ delete ‘a’

Authors: We deleted ‘a’ and highlighted the change.

Authors: We fixed an additional mistake in the paragraph starting with ‘To address the disparity of explanatory …’. We switched ‘positive’ and ‘negative’ words and highlighted them in the text. 

Authors: We also removed the sentence ‘We denoted the number of positive and negative data points with ‘P’ and ‘N’, respectively.’ From the end of the paragraph starting with ‘In each round, we calculated sensitivity …’. We did not use the ‘P’ and ‘N’ variables in the study. 

Prediction:

• Pg 13 Suggest change ‘Due to high imbalanced nature of the data set’ to ‘Due to the highly imbalanced nature’

Authors: We changed the sentence accordingly and highlighted it in the text.

Results and discussion:

• Pg 16 delete ‘as well’ after Belkhiria et al., 2018)

Authors: We deleted ‘as well’. Thanks. 

Conclusion and future work:

• ‘comprehensive list’ – same comment as before

Authors: We changed it to ‘extensive’ in abstract, the first paragraph of Motivations and the first paragraph of Conclusions.

• In the methodology section it states: ‘In the last step (Prediction), end-users can communicate with the system through a user interface. Here, the user interface can include mapping and monitoring of the risk, given a current spatio-temporal dataset.’ Where is the link to the user interface? Have any maps been generated from this work?

Authors: The results from this work in conjunction with a social media disease surveillance study was used to build a decision support system in a subsequent study that recently was accepted to be published. In the above-mentioned study, we explained how the user interact with the system and risk maps are generated. 

References:

• Please check references for H and N rather than h and n.

Authors: We checked and capitalized the strain names in the reference.

---

## [Decision Letter · Decision Letter 1]

17 Dec 2020

PONE-D-20-03935R1

A Framework for the Risk Prediction of Avian Influenza Occurrence: An Indonesian Case Study

PLOS ONE

Dear Dr. Dara,

Thank you for submitting your manuscript to PLOS ONE. After careful consideration, we feel that it has merit but does not fully meet PLOS ONE’s publication criteria as it currently stands. Therefore, we invite you to submit a revised version of the manuscript that addresses the points raised during the review process.

One of the two reviewers is satisfied with your revision. Another reviewer, on the other hand, deems the technical content as adequate, but requires you to clarify some aspects with respect to data treatment. I concur with the reviewer that such clarifications will contribute to make the paper more readable and to significantly increase its visibility and impact. 

We look forward to receiving your revised manuscript.

Kind regards,

Alessandro Rizzo

Academic Editor

PLOS ONE

Reviewers' comments:

Reviewer's Responses to Questions

**Comments to the Author**

1. If the authors have adequately addressed your comments raised in a previous round of review and you feel that this manuscript is now acceptable for publication, you may indicate that here to bypass the “Comments to the Author” section, enter your conflict of interest statement in the “Confidential to Editor” section, and submit your "Accept" recommendation.

Reviewer #1: All comments have been addressed

Reviewer #2: All comments have been addressed

2. Is the manuscript technically sound, and do the data support the conclusions?

Reviewer #1: Yes

Reviewer #2: (No Response)

3. Has the statistical analysis been performed appropriately and rigorously? 

Reviewer #1: Yes

Reviewer #2: (No Response)

4. Have the authors made all data underlying the findings in their manuscript fully available?

Reviewer #1: Yes

Reviewer #2: (No Response)

5. Is the manuscript presented in an intelligible fashion and written in standard English?

Reviewer #1: Yes

Reviewer #2: (No Response)

6. Review Comments to the Author

Reviewer #1: Even if authors answered to revision issues, my personal problem with this work is still related to the partially addressed evidence and focus on geographic data aspects. The answer to revision is indeed not so clear with respect to the reviewer invitation on GIS related aspects (i.e. lat and long information at the first revision step). Spatial modules and images are strictly related to epidemiology performance and analysis. Thus, even if authors addressed the reviewer comments they did not mentioned the technical aspects related to mapping gathered data in a native spatial system. Now, if this is not an issue that should be deeply treated, it is necessary in case of: (i) integration of different sources and (ii) in case of reuse of the proposed method for different cases. How data has been mapped into commonly used and structured geographic model, is still not clear. Performing interesting queries as well as epidemiological analysis is what reader should expect from this paper starting from examples, title and abstract. On the other hand, if the proposal is finalized as a presentation of a use case my suggestion is to improve the data structure representation as minor revision to allow readers in gathering the contribution as a well structured example. I thus suggest to review the methodology to enrich possibility of replicating on different data samples.

Reviewer #2: (No Response)

7. PLOS authors have the option to publish the peer review history of their article (what does this mean?). If published, this will include your full peer review and any attached files.

Reviewer #1: No

Reviewer #2: No

---

## [Author Response · Author response to Decision Letter 1]

18 Dec 2020

Date: 17 Dec 2020

Title: A Framework for Risk Assessment of Avian Influenza Occurrence: An Indonesian Case Study

Authors: Samira YousefiNaghani, Rozita Dara, Zvonimir Poljak, Fei Song, and Shayan Sharif

Manuscript No: PONE-D-20-03935R1

Dear Dr. Alessandro Rizzo

We are very grateful to Academic Editor and the reviewers for the second review and comments regarding the Manuscript ID PONE-D-20-03935R1 entitled "A Framework for Risk Assessment of Avian Influenza Occurrence: An Indonesian Case Study". 

We have addressed the comments and made the required changes which we believe have improved the manuscript in a way that is now acceptable for publication in PLOS ONE journal. In the following, we will describe our responses to the points made by academic editor and reviewers. Given comments have been copied and are followed by specific responses in green. 

Academic Editor Comments:

• One of the two reviewers is satisfied with your revision. Another reviewer, on the other hand, deems the technical content as adequate, but requires you to clarify some aspects with respect to data treatment. I concur with the reviewer that such clarifications will contribute to make the paper more readable and to significantly increase its visibility and impact. 

Authors:

• We added a paragraph in the Methodology section to add more clarity on data aggregation. The paragraph is highlighted in the manuscript. For the purpose of reusability, we can provide the data to be publicly available for readers.

Authors:

• We uploaded the EPS figures on PACE tool. These figures have been adjusted and generated TIFFs figures. We will be submitting TIFF figures along with the revised manuscript. 

• In addition, we renamed the figures to their associated caption names. 

Reviewer #1 comments:

• Even if authors answered to revision issues, my personal problem with this work is still related to the partially addressed evidence and focus on geographic data aspects. The answer to revision is indeed not so clear with respect to the reviewer invitation on GIS related aspects (i.e. lat and long information at the first revision step). Spatial modules and images are strictly related to epidemiology performance and analysis. Thus, even if authors addressed the reviewer comments they did not mentioned the technical aspects related to mapping gathered data in a native spatial system. Now, if this is not an issue that should be deeply treated, it is necessary in case of: (i) integration of different sources and (ii) in case of reuse of the proposed method for different cases. How data has been mapped into commonly used and structured geographic model, is still not clear. Performing interesting queries as well as epidemiological analysis is what reader should expect from this paper starting from examples, title and abstract. On the other hand, if the proposal is finalized as a presentation of a use case my suggestion is to improve the data structure representation as minor revision to allow readers in gathering the contribution as a well structured example. I thus suggest to review the methodology to enrich possibility of replicating on different data samples.

Authors:

• Thanks for your helpful suggestion. On page 6, in the Methodology Section and after Fig 3, we added a paragraph explaining how the relational dataset was used as a geographical model which can be visualized in a GIS system. We added the paragraph below:

``The data aggregation was performed in a Structured Query Language table. The response variable describes a disease event that occurred within a certain time interval (week number t) and a spatial compartment with a center of (x,y), where x represents longitude and y represents latitude. The spatiotemporal information in ``Indonesia" data tables in Fig 2 are demonstrated by ``week", ``cell_longitude" and ``cell_latitude" attributes. Similarly, for other covariates, pre-processing techniques were applied to transform the records into the defined grid-based table. Since the table included coordinate information for each record, a weekly timeline of information could be visualized in maps using GIS software. Moreover, extracted patterns from the grid-based data could be simple and easy to understand.”

---

## [Decision Letter · Decision Letter 2]

23 Dec 2020

A Framework for the Risk Prediction of Avian Influenza Occurrence: An Indonesian Case Study

PONE-D-20-03935R2

Dear Dr. Dara,

We’re pleased to inform you that your manuscript has been judged scientifically suitable for publication and will be formally accepted for publication once it meets all outstanding technical requirements.

Kind regards,

Alessandro Rizzo

Academic Editor

PLOS ONE

Additional Editor Comments (optional):

As per the reviewer suggestion, and considering the authors' availability, I recommend the authors to make their database public and add a reference to the dataset in the final version of the manuscript.

Reviewers' comments:

Reviewer's Responses to Questions

**Comments to the Author**

1. If the authors have adequately addressed your comments raised in a previous round of review and you feel that this manuscript is now acceptable for publication, you may indicate that here to bypass the “Comments to the Author” section, enter your conflict of interest statement in the “Confidential to Editor” section, and submit your "Accept" recommendation.

Reviewer #1: All comments have been addressed

2. Is the manuscript technically sound, and do the data support the conclusions?

Reviewer #1: Yes

3. Has the statistical analysis been performed appropriately and rigorously? 

Reviewer #1: Yes

4. Have the authors made all data underlying the findings in their manuscript fully available?

Reviewer #1: No

5. Is the manuscript presented in an intelligible fashion and written in standard English?

Reviewer #1: Yes

6. Review Comments to the Author

Reviewer #1: Even if authors reply to reviewer in a quite simple way (adding one paragraph), authors claim that they may put available datasets for GIS applications or to similar studies. My suggestion is to accept author proposals that may make available dataset adding also a reference to their manuscript.

7. PLOS authors have the option to publish the peer review history of their article (what does this mean?). If published, this will include your full peer review and any attached files.

Reviewer #1: No

---

## [Editor Report · Acceptance letter]

6 Jan 2021

PONE-D-20-03935R2 

A framework for the risk prediction of avian influenza occurrence: an Indonesian case study 

Dear Dr. Dara:

I'm pleased to inform you that your manuscript has been deemed suitable for publication in PLOS ONE. Congratulations! Your manuscript is now with our production department. 

Kind regards, 

on behalf of

Prof. Alessandro Rizzo 

Academic Editor

PLOS ONE